# A Note on the Reproducibility of Chaos Simulation [note 1]

**DOI:** 10.3390/e22090953

**Published:** 2020-08-29

**Authors:** Thalita E. Nazaré, Erivelton G. Nepomuceno, Samir A. M. Martins, Denis N. Butusov

**Affiliations:** 1Control and Modelling Group (GCOM), Department of Electrical Engineering, Federal University of São João del-Rei, São João del-Rei, MG 36307-352, Brazil; thalitanazare@gmail.com (T.E.N.); martins@ufsj.edu.br (S.A.M.M.); 2Youth Research Institute, Saint-Petersburg Electrotechnical University “LETI”, 197376 Saint Petersburg, Russia; dnbutusov@etu.ru

**Keywords:** reproducibility, computational chaos, computer arithmetic, OSF platform

## Abstract

An evergreen scientific feature is the ability for scientific works to be reproduced. Since chaotic systems are so hard to understand analytically, numerical simulations assume a key role in their investigation. Such simulations have been considered as reproducible in many works. However, few studies have focused on the effects of the finite precision of computers on the simulation reproducibility of chaotic systems; moreover, code sharing and details on how to reproduce simulation results are not present in many investigations. In this work, a case study of reproducibility is presented in the simulation of a chaotic jerk circuit, using the software LTspice. We also employ the OSF platform to share the project associated with this paper. Tests performed with LTspice XVII on four different computers show the difficulties of simulation reproducibility by this software. We compare these results with experimental data using a normalised root mean square error in order to identify the computer with the highest prediction horizon. We also calculate the entropy of the signals to check differences among computer simulations and the practical experiment. The methodology developed is efficient in identifying the computer with better performance, which allows applying it to other cases in the literature. This investigation is fully described and available on the OSF platform.

## 1. Introduction

One of the essential aspects of science is the ability to reproduce scientific results reported by other scientists [1]. This perspective has been the subject of discussion in the field of control theory and automation in several aspects. For example, Bonsignorio [2] stated that the reproducibility of experimental results is a key feature of the scientific method. Despite this, the area of robotics and artificial intelligence has not prioritized this perspective. Similarly, the IEEE panel of editors led by IEEE Robotics and Automation Magazine’s Editor-in-Chief stated that reproducibility in scientific research is imperative, but that unfortunately, there are many scientific papers for which reproducing results is difficult or even impossible [3]. In control theory, the relevance of system modelling and simulation is widely recognized [4].

Many researchers are confident in relating chaos theory with electronic circuits, developing more and more new models of components and systems that may present chaotic behaviour, as for example, in the recent articles [5,6,7], in which the authors proposed models of systems and circuits applicable in different areas. Since chaotic systems are so hard to understand analytically, numerical simulations assume a key role in their investigation [8]. Such simulations have been considered as reproducible in many works. Despite this recognition, less attention has been paid to the effects of computer finite precision on the reproducibility of nonlinear dynamical systems’ simulation [9]. In fact, numerical experiments have been performed since Lorenz’s works [10] with the intention of understanding the behaviour of nonlinear dynamical systems [11,12].

Thus, it is possible to state that numerical computation is of fundamental importance in the analysis of electronic circuits that describe nonlinear dynamic systems [13,14]. These analyses make use of popular software and computers that are easily accessible to most researchers. However, there are many published studies whose reliability of their numerical results has not been carefully evaluated [15]. Among the studies found in the literature that question the reliability of numerical simulations, especially those involving chaotic dynamic systems, there is [16], wherein the authors presented interval computation as a way to obtain more reliable results in the simulation of the logistic map. Qin and Liao [17] showed the strong influence of the rounding error on the reliability of numerical simulations of chaotic dynamical systems. The work carried out by Teixeira et al. [18] pointed out the problem of the numerical convergence of chaotic systems and that this convergence cannot always be guaranteed. Nazaré et al. [19] presented the influence of the initial condition and the numerical inaccuracy on the logistic and quadratic map simulation result, respectively. Nepomuceno and Martins [20], Nepomuceno et al. [21] demonstrated the lower bound error theorem to estimate the error in recursive functions, such as the NARMAXpolynomial models. This finding has been proven to be useful for testing the reliability of numerical simulations for nonlinear dynamical systems.

Programming languages, such as MATLAB, C, Fortran, Octave or Python, are among the most used software to investigate chaos simulation [22,23]. Works that use SPICE-type software are less present, but they are of great importance, particularly for electronic circuits’ investigation [24]. In general, there is a lack of important details such as the software version and computer settings with which the simulations were performed. This fact hinders the future analysis of the same circuit, if it does not have good reproducibility. As an example, there was the work done by Šalamon and Dogša [25], in which the authors identified problems reproducing simulations of Chua’s circuit on different computers using Multisim software. There is also the work of Milani et al. [26], which presented the influence of software and operating systems on simulations of nonlinear dynamic models.

In this study, we investigate the reproducibility of the chaos simulation of the jerk circuit proposed by Sprott [14]. Simulation data obtained by different computers using LTspice XVII software are compared with results collected experimentally. This comparison is performed using the Normalised Root Mean Squared Error (NRMSE) index calculation. The objective of this index is to identify the computer that has the largest prediction horizon. We also calculate the entropy of the signals to check differences among computer simulations and the practical experiment. The proposed technique is efficient in identifying the computer with better performance. Supplementary material is available on the OSF platform to facilitate the reproduction of this work.

The remainder of this paper is organized as follows. Section 2 presents the basic concepts of the article. Section 3 presents some related works, while Section 4 shows the methodology covered in the paper. Then, in Section 5, the results obtained are described and analysed. Finally, Section 6 presents the discussion and perspectives of future work.

## 2. Preliminary Concepts

### 2.1. Jerk Dynamics

Among the circuits that present chaotic behaviour, it is possible to find a model that can be described by the system of differential equations of order three. The solution of this equation can be found in [27]. The nonlinear function *J* is called “jerk”; it describes the third-order derivative of *x*, which corresponds to the first derivative of acceleration in a mechanical system [14]. This system is described as follows:(1)$$\dddot{x}=J\left(\ddot{x},\dot{x},x\right),$$
where the *J* function is given by $J=-A\ddot{x}-x\mp {\dot{x}}^{2},$ which has been considered as the simplest ordinary differential equation with a quadratic nonlinearity that has a chaotic result. The nonlinear term of *J* is $\dot{x}$. In this equation, *A* is the bifurcation parameter that presents chaos in the range of 2.0168<A<2.0577 [14].

### 2.2. Normalised Root Mean Squared Error

The NRMSE index can be calculated as follows:$$NRMSE=\frac{\sqrt{\sum_{k=1}^{N}{\left[y\left(k\right)-\hat{y}\left(k\right)\right]}^{2}}}{\sqrt{\sum_{k=1}^{N}{\left[y\left(k\right)-\bar{y}\right]}^{2}}},$$
where *y* is the collected data, $\hat{y}$ is the simulated data, $\bar{y}$ is the average of the simulated data and *N* is the amount of points analysed.

It is well known that chaotic orbits diverge exponentially by a slight difference in the initial conditions. Strict use of the NRMSE is useless if no care for this issue is considered. Here, we are interested in comparing the growth of NRMSE when equal initial conditions are considered in simulations occurring in different computers. In such a way, we believe that this index would offer a measure of reproducibility for specific hardware.

### 2.3. Entropy Analysis

Shannon [28] proposed a mathematical method for measuring how much information is in the transmission of a signal. The Shannon entropy is calculated according to Equation (Equation 2):(2)$$H\left(x\right)=\sum P_{i}log_{2}\frac{1}{P_{i}}.$$

Thus, the entropy calculation can be used to compute the statistical measure of the uncertainty of a given signal, based on the probabilities of information representation. As Woodward [29] said, with the method proposed by Shannon to calculate entropy, the fabrication of a message technically becomes an act of selection, just as a composer selects lyrics, a musician selects notes and a painter selects colours. The signal has been changed into the “uint8” format. The maximum value of 8 using a base-2 logarithm is accomplished for a uniform random signal.

### 2.4. Open Research Culture

The importance of an open research culture is generally accepted among scholars. Over the past few years, important journals such as Nature [1,30,31,32,33,34,35] and Science [36,37,38,39,40,41,42,43,44,45,46] have published papers in such a direction. Transparency, openness and reproducibility are generally recognized as essential features of science. Nevertheless, published papers reveal a different practice [44]. In fact, the same relevance seems to be on a different level for other journals. For instance, Entropy magazine covers topics such as chaos, nonlinear dynamics, information transfer and information loss, which can be easily related to this work.

One of the best initiatives to adopt transparency and openness in science was made by the Transparency and Openness Promotion Committee (Charlottesville, Virginia, USA). Figure 1 summarises the eight standards and three levels of the guidelines organized by this committee. A key point of this standard is the involvement of authors and publishers in all stages of a science project. Furthermore, in the seminal paper proposed by this team [44], they recommend the OSF platform, a free open platform to support research and enable collaboration (See more details at https://osf.io/). Besides the collaboration, this platform offers a very intuitive way to describe research projects and gives hints on how to reproduce the results. The project is addressed permanently by a DOI link.

## 3. Related Works

Chaos theory has been studied and used by several authors over time, and it is possible to find its application in different areas; for example, chaotic oscillators as a form of inductive sensors [47] or analysing the influence of external effects, such as temperature, on chaotic circuits physically implemented [48]. The study of chaos is also widely applied in the analysis of the effect of arithmetic computation in simulations of nonlinear dynamic systems, in which it is possible to find works such as those by [49], in which the authors analysed aspects of computational arithmetic, and their finite precision, in three types of dynamic systems. In [50], the main focus was on digital chaos, that is chaotic systems analysed in computational environments, and the influence of the finite precision of computers on the dynamic properties of the system, since these properties change completely when compared to continuous analysis. Sayed et al. [51] discussed the sensitivity of implementing chaotic systems and the effect on the incompatibility between time series implemented in double precision software and single precision floating point and fixed point hardware, and the authors used this incompatibility signal for the generation of pseudo-random numbers and application in image encryption.

Chaotic systems are also widely used as Pseudo-Random Number Generators (PRNGs). Persohn and Povinelli [52] related PRNGs to the finite precision of computers by means of an investigation of digital chaos and verified the influence of finite precision on PRNGs. Sayed et al. [53] developed a PRNG in fixed-point hardware, using the logistic map that experiences a trade-off between computational efficiency and precision. These generators are widely used in the area of cryptography, as stated by the authors of [54], who also investigated the limitation of the computer, which impacts the dynamic degradation of the chaotic properties of the system. In that work, the authors explored this limitation of the computer as a source of randomness instead of providing this characteristic. On the other hand, in [55], the authors sought to provide this characteristic, proposing a method of chaotification to solve the problem of the degradation of the dynamics of chaotic circuits. In [56], the authors aimed at investigating the properties of the tent map and analysed the effect of the implementation of fixed point precision on the periodicity of a single map, and later, they also analysed the effect on the statistical properties of the generated sequence of coupled tent maps, showing that the results obtained have acceptable random properties and are suitable for cryptographic applications.

It is possible to find at least 35 works [57,58,59,60,61,62,63,64,65,66,67,68,69,70,71,72,73,74,75,76,77,78,79,80,81,82,83,84,85,86,87,88,89,90,91] in the Entropy journal that relate chaos theory in different areas; however, there are only five articles, from all areas, where reproducibility was mentioned [92,93,94,95,96]. From those aforementioned papers, only Funabashi [95] presented some slight relationship with the reproducibility of nonlinear dynamics fields. In that paper, the author defined a conceptual model that incorporates the quality of observation in terms of accuracy and reproducibility; he devoted Section 2 to setting up scientific measures that should assure reproducibility. This notwithstanding, the principles are devoted to building models for a specific case of interactive data-driven citizen science, and there are no standard guidelines and involvement with the community of journals and scientists. Although in the literature, the concern for the reproducibility of results is growing and there is a considerable number of articles that address chaos and loss of information, it is still not possible to find in Entropy any work that directly links these two themes.

Apart from what has been said, we also present some related investigations on chaos theory and highlight some limitations on their reproducibility. The authors of [81] investigated a new model for a voltage-controlled memristor, called an absolute memristor, because it has an absolute value term. They used Multisim software to simulate the circuit model based on the memristor. The circuit presented by them can be used as a pseudo-random number generator to provide key sequences for digital encryption systems. Figure 2 shows the results obtained using MATLAB software and the SPICE Multisim simulator. At the end of the work, the authors stated:

The simulation results of the chaotic circuit obtained by the Multisim software matched well with the numerical simulation results obtained by the MATLAB software. The results of the DSP experiment and the NIST test indicated that the proposed chaotic system can be used as a pseudo-random sequence generator to provide key sequences for encryption systems. Therefore, the proposed chaotic system can be efficiently applied for digital information encryption.

It is a fact that digital cryptography is a topic that is being increasingly addressed by the scientific community. Researchers are increasingly interested in developing systems and circuits that can generate random keys to make information more secure, and chaotic systems are proposed by them to further enhance security. The reproducibility of such results is essential. As we have seen, in this article, the authors presented a circuit to generate the encryption key, using SPICE software. If this key is generated on one computer and later it is necessary to reproduce it on another one with a different configuration, it is likely that this will not be possible. This observation emphasises even more the importance of having works in the literature that deal with reproducibility.

The last work to be presented in this section was devoted to developing a new model of a memcapacitor and an emulator for their circuit. A chaotic oscillator was designed, and they analysed the characteristics and dynamic behaviour of the system and further reinforced: “both analytically and experimentally”. Again, the authors provided only the details of the system for the experiments and the software used in the simulation. They did not make any comparisons of the experimental and computational results; making it even more difficult to reproduce the work. To conclude the work, the authors stated:

Moreover, the new memcapacitor-based system was realised using an experimental circuit, which agreed well with the numerical simulations and verified the theoretical analysis results.

It is possible to notice that they succeeded in the proposed methodology, wherein the simulation and experimental results were in good agreement. Nevertheless, it is undoubted that there is room to increase the rigour in such a comparison.

Note that in all the mentioned papers [60,71,73,81,90], the authors related simulations to experimental data to validate a proposed model. In all cases where physical circuits were built, they described the components and indicated their configurations. In some of the works, it is possible to find information about the simulation software, and only one of them indicated which version was used. However, in none of the studies was rigorous care taken regarding the reproducibility of the simulation results. It is clear that chaotic systems and circuits can and are being used more and more for different applications. One of the areas that has been gaining more prominence is in cryptography, and especially in this application, the attention regarding the reproducibility of the results must be even greater.

## 4. Methodology

As mentioned, this study is aimed at reproducing the simulation and implementation of the electronic circuit proposed in the article. This simulation was performed on different operating systems and processors, allowing us to analyse the results obtained by the practical implementation and on each computer, thus checking the reproducibility of LTspice. In this work, we adopted the integration method “modified trap” with standard parameters of LTspice, which allows a non-fixed step in the simulation.

Reproducibility is understood as the possibility to reproduce the same or similar simulation results when subjected to the same initial conditions, circuit parameters and numerical method configuration parameters. Thus, the methodology adopted to perform these procedures can be summarized in the following stages:(1)Practical implementation of an electronic circuit.(2)Collection of implementation data via a data acquisition board.(3)Simulation and data collection of the electronic circuit scheme in different computers as described in Table 1. The implementation in a board is shown in Figure 3.(4)Calculation of the NRMSE index for choosing the reference computer and analysis of the reproducibility between the signals obtained on the computers.(5)Calculation of entropy to verify the similarity among all data collected in simulations and the practical experiment.

It is important to stress that other indexes could be used, such as the Lyapunov exponent, correlation dimension, partial dimension or mutual information [97]. Nevertheless, NRMSE and entropy are suitable and computationally low cost for our purpose.

To ensure that all circuit simulations start from the same initial condition as the experimental circuit, first, the practical implementation of the circuit was performed; thus, the first value collected in practice was used as an initial condition for all simulations.

After performing these procedures, the comparison between simulated and practical results was made by the NRMSE index. The reason for choosing this index to perform the comparison between the results obtained on the 4 computers was due to the simplicity of its use, which can be calculated in any software. They are mathematical operations that assist in the analysis of reproducibility between two sets of data, being able to manipulate this set and check the behaviour of the signals during the entire time interval or divided into smaller intervals.

The comparison between simulation and experimental data is not straightforward because the length of the vectors is different. To overcome this problem, simulated and experimental data were stored in the form of vectors, and for the calculation to present minimal error, these vectors needed to be of equal length. Apart from data, initialisation mode and the number of points collected by LTspice presented the differences of one computer from another, resulting in differently sized voltage vectors for each computer tested. As a solution to this problem, in the routine implemented for the index calculation, a new time vector with a previously established number of points was created, so that the interpolation of the original vectors could be performed. By interpolating all the voltage vectors collected computationally and in practice, we now have the same number of points to be analysed.

The computer simulation that presented the lowest NRMSE was considered as a reference. Then, using this computer as a reference, a comparison was made with the other simulations to verify the reproducibility of the results between the computers, which is the main objective of this work. Entropy calculation, as presented in Equation (Equation 2), was used as another method of verifying the compatibility of information transmitted by computers and the physical circuit. Regarding the error propagation, care was taken to follow the guidelines contained in the standard of the Institute of Electrical and Electronic Engineering (IEEE) 754–2019, which deals with floating-point computation [98,99], and the IEEE standard on Interval Arithmetic, IEEE 1788–2015 [100].

## 5. Results

The jerk circuit was simulated in LTspice XVII on the four computers listed in Table 2. The same version of the software was used in all computers. Then, the data from the physical circuit were collected using the data acquisition board. As mentioned earlier, to ensure that all simulations on the circuit started from the same initial condition as the experimental circuit, the first value obtained in practice was used, so the initial condition for all simulations was $\left[\ddot{x},\dot{x},x\right]=\left[-4.604280\times 10^{-1},0,0\right]$. All results were collected in the time interval within 0 and 0.1 s. The signal collected in the practical experiment is shown in Figure 5, while Figure 6 presents the response to voltage collected at the point $\ddot{x}$ for each computer.

Results were compared to verify the reproducibility among chaos simulation in different computers, as shown in Table 2. The comparison was performed using a MATLAB software routine to calculate the NRMSE index considering all the interpolated voltage values of each computer. The results obtained are presented by Table 1. Analysing only the graph presented in Figure 6, it is believed that there is no variety in the results from one computer to the other. However, the NRMSE index reveals some differences.

As we previously mentioned, the computer that presented the lowest NRMSE value was considered as a reference. To further investigate the performance of this computer, the calculations were split into ten windows of 4700 points. The index calculation was carried out for each of these windows. It can be observed by Table 3 that with the increase in the number of points collected, there is also the increase of the index value, as expected. Computers 1 and 4 are the ones with the lowest values. However, for comparison of the last part where all points are analysed, Computer 4 performs better. Thus, it was chosen as a reference to verify the reproducibility of LTspice.

The comparison between the lowest NMRSE computer and experimental data is presented in Figure 7. In the first moments, the voltage values remain close between the reference Computer 4 and the experimental data. Thus, it is possible to observe that the longer the simulation period, the greater the difference between the results of each computer is.

After recalculating the NRMSE between computers using Computer 4 as a reference, the values shown by Table 4 are obtained. With these results, it is possible to demonstrate that the reproducibility of the software is higher when the characteristics of the computers are similar. It can also be observed that Computer 2 presents the most different result; it is also the one that presents the lowest reproducibility. However, even resulting in small values, it is noted that there is a difference in response between two computers with similar configurations. One can see this difference better looking at Figure 2, which presents a comparison between the voltage values obtained for each computer. It can be seen that Computer 1 virtually follows the entire simulation range of Computer 4 since the NRMSE index value is close to zero. On the other hand, Computer 2 at the beginning of the simulation presents a considerable difference compared to other computers.

As previously mentioned, the entropy of the signals was also calculated to verify the compatibility of the information transmitted by the computers and by the practical experiment. The result found can be seen in Table 5. With these values, it is possible to verify that the reproducibility indication found with the calculation of the NRMSE index is confirmed with the calculation of entropy. The entropy value of the signal collected on Computer 4 is the closest to the value for the data collected in practice. The same comparison is valid for reproducibility between computers. With this result, it can be said that Computer 1 is the most capable of transmitting a signal similar to Computer 4. In contrast, the entropy value of Computer 2 is the most distant from Computer 4.

To make a more improved analysis and better check the reproducibility of chaotic circuit simulations in SPICE software, the same jerk circuit was simulated, with the same conditions in two different versions of the LTspice software. It is important to note that the simulations took place on the same computer. The versions used were LTspice IV and LTspice XVII, and the simulations occurred in the time interval from 0 to 0.5 s. After collecting data from both software, the NRMSE index was calculated again, which can be seen in Table 6. Figure 8 shows the simulations of the two versions. In Figure 8a, we have the entire simulated time interval, and it is already possible to see the difference between the two simulations. Figure 8b shows the results only from the first moments. In Figure 8c, we have a time interval in which the results differ considerably, thus reinforcing the influence of the simulation time on the reproducibility of the results. It is possible to perceive that even simulating on the same computer, the fact of using different versions of the software also influences the reproducibility of the results. Sayed and Fahmy [101] also reported this problem when simulating the same system in different software.

At the end of all analyses, it was possible to observe the same reproducibility problem found by [25] when trying to reproduce the Chua circuit using the Multisim software. We noticed that the simulation jerk circuit using the same version of LTspice software on different computers or different software versions produced different outcomes.

In addition to verifying the difference in reproducibility between computers and software versions, with the proposed methodology, it was possible to observe the influence of the simulated time interval. In all results presented, the NRMSE index varied according to the time chosen for analysis. Even for Computers 1 and 4, which have good reproducibility, it can be said, based on the other comparisons, that if the simulation time were longer, this index would tend to increase.

### Brief Presentation of OSF

In this section, we briefly present the project of this manuscript on the OSF platform. We can access the project by means of:DOI: https://doi.org/10.17605/osf.io/jhtpdShort URL: https://osf.io/jhtpd/.

Figure 9 shows the opening screen for the project on OSF. We indicate by red capital letters some main features of this platform.

(A)The title of the project.(B)The project’s authors.(C)Permanent link of the project (DOI).(D)A Wiki description of the project.(E)All files related to the project. The reader can download all files in a unique zip file.(F)Latex source of the file. In this case, we used a storage add-on to connect with Dropbox, which is linked to Overleaf, where this manuscript was edited.(G)Code files used to simulate and present the results of the paper.(H)An illustrative example of the project.(I)All images used in the project.(J)Simulations files of LTspice. There is also a specific library created to simulate the components of the circuits.(K)This indicates the branch version of the project. The zero means that it is in its original version.(L)How to cite this project.(M)Recent activity made by the contributors. It is possible to notice that the last activity was a change in the title of the project (manuscript) from “A note on the reproducibility of chaotic systems simulation” to “A note on the reproducibility of chaos simulation”.

## 6. Discussion

We investigated the reproducibility of the chaos simulation of an electronic circuit proposed by [14]. The simulation was carried out in LTspice over four different computer hardware and operating systems. We proposed the use of the NRMSE index and the calculation of entropy to evaluate the level of reproducibility of each computer. Experimental data were used to choose the best computer from the point of view of horizon prediction. We also suggested using the OSF platform as a systematic way to share code, files and technical details essential to reproduce the results in this paper.

The contribution of this paper is twofold. First, we again evidenced, but for a different chaotic system and software simulation, the results shown by [25]. It was not possible to reproduce the chaos simulation of the jerk circuit using LTspice without the knowledge of the operational system and computer hardware. This can also be observed when analysing the entropy values for each signal collected. Different values for entropy reinforce the conclusion that computers may produce different simulation outcomes, even with the same software. Another important result that was presented is the fact that simulating the same circuit on the same computer, but using two different versions of the LTspice software, the results were not precisely reproducible. With all the results of the NRMSE index, it was also possible to see that the simulation period has a strong influence on the analysis of the reproducibility of the results. Additionally, the NRMSE index was useful to evaluate the reproducibility level of each computer. Experimental data were used to choose a computer as a reference. From this study, it was suggested that similar computer hardware and operating systems (Computers 4 and 1) present closer results. Second, the use of the OSF platform was presented as a way to present all possible necessary details to reproduce the chaos simulation.

Far from this work presenting a definitive solution to the problem, it is believed that the methodology employed is able to identify the problem and choose computers with better performance in terms of the horizon of prediction. The IEEE Code Ocean is a cloud platform where researchers and readers of scientific work execute the code on the same hardware and software [102]. It is an interesting initiative to keep track of the simulation. Nevertheless, its focus is only on language programs such as MATLAB, Octave, R and Python. It is clear that there is much less attention given to repositories for graphical code, such as Labview and LTspice. Certainly, it brings a reproducibility perspective that is vastly superior to what has been presented in most literature so far and deserves the attention of the scientific community. Additionally, as a perspective of future work, we intend to propose a method using the Lyapunov exponent simulation, which indicates the time interval for reliable results. This can be of great interest because even in Code Ocean, different languages can offer different results.

## Figures and Tables

**Figure 1 entropy-22-00953-f001:**
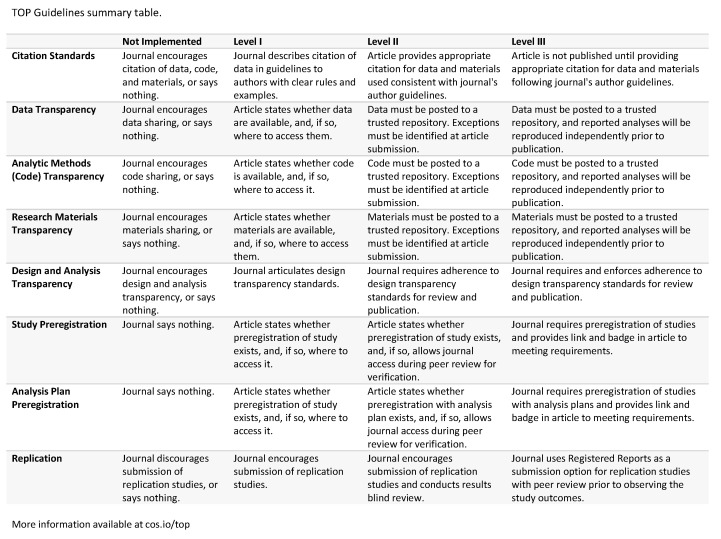
Summary of the eight standards and three levels of the guidelines organized by the Transparency and Openness Promotion (TOP) Committee meeting at the Center for Open Science in Charlottesville, Virginia, USA, in 2014 [44]. This guideline is Version 3 of 2018 available at http://cos.io/top.

**Figure 2 entropy-22-00953-f002:**
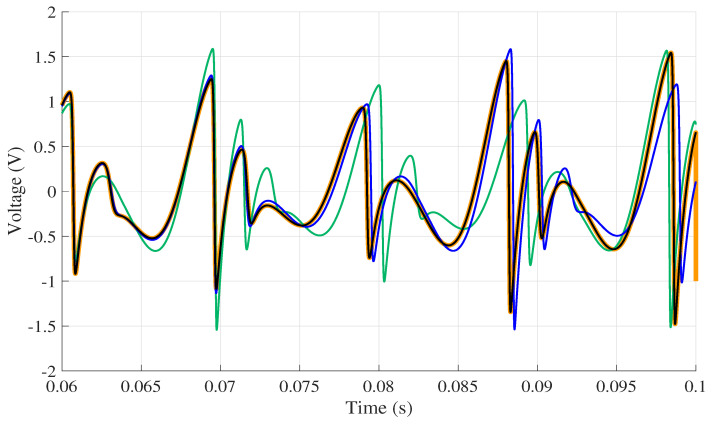
Voltage in $\ddot{x}$ for each computer analysed. (
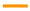
) Computer 1; (

) Computer 2; (

) Computer 3; (
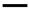
) Computer 4.

**Figure 3 entropy-22-00953-f003:**
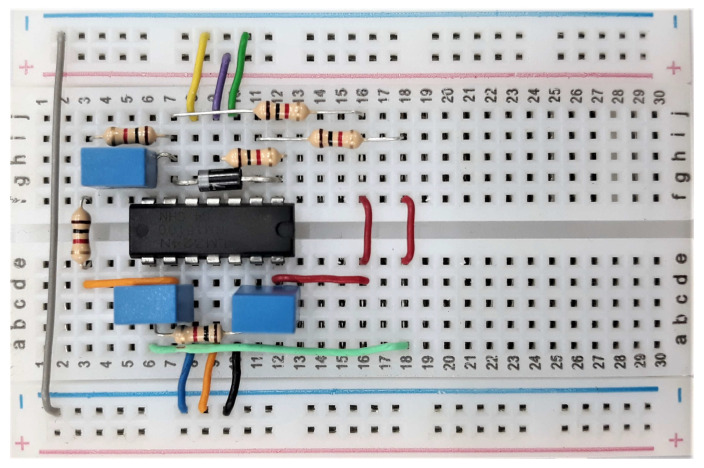
Implementation of the jerk circuit scheme (Figure 4). LM 324 replaces the original Tektronix AM 501 operational modules, since this device consists of four independent high-gain frequency-compensated operational amplifiers.

**Figure 4 entropy-22-00953-f004:**
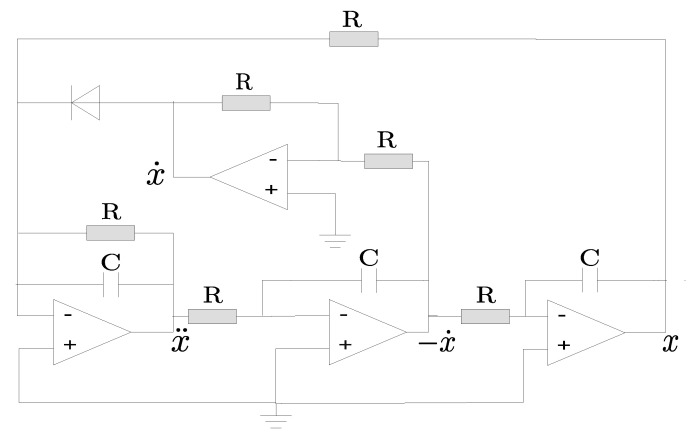
Chaotic circuit proposed by [14]. The components used are a resistor *R* of 1 kΩ, operational amplifiers (LM324), 1 μF capacitors and the 1N4001 diode.

**Figure 5 entropy-22-00953-f005:**
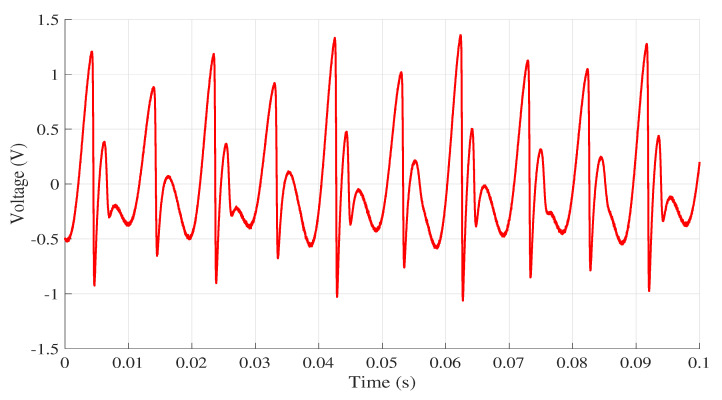
Voltage collected from the practical implementation of the circuit.

**Figure 6 entropy-22-00953-f006:**
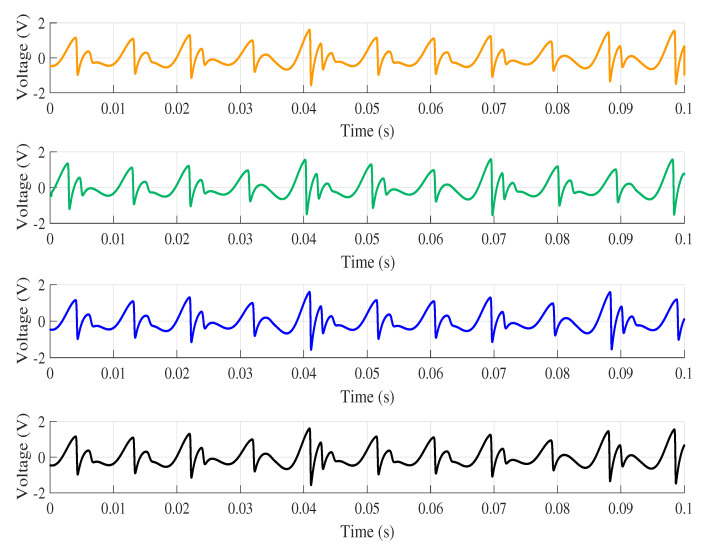
Voltage collected at point $\ddot{x}$ of each computer analysed. The legend is as follows. (
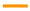
) Computer 1; (

) Computer 2; (

) Computer 3; (
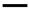
) Computer 4.

**Figure 7 entropy-22-00953-f007:**
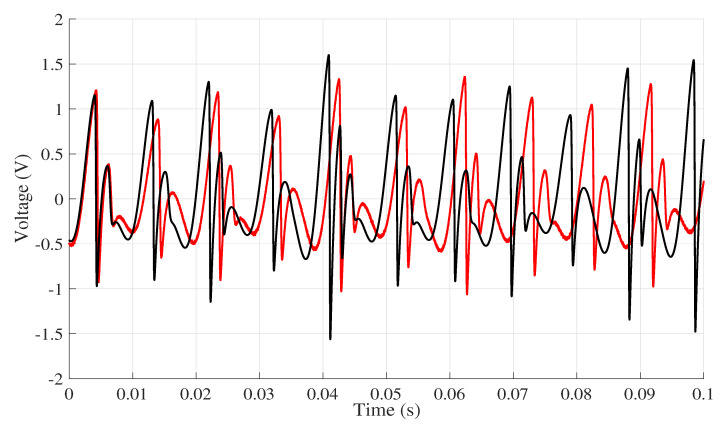
Voltage in $\ddot{x}$ for Computer 4, which presented the lowest NRMSE index, and the data collected experimentally. The legend is as follows. (
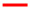
) Experimental data and (
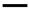
) Computer 4.

**Figure 8 entropy-22-00953-f008:**
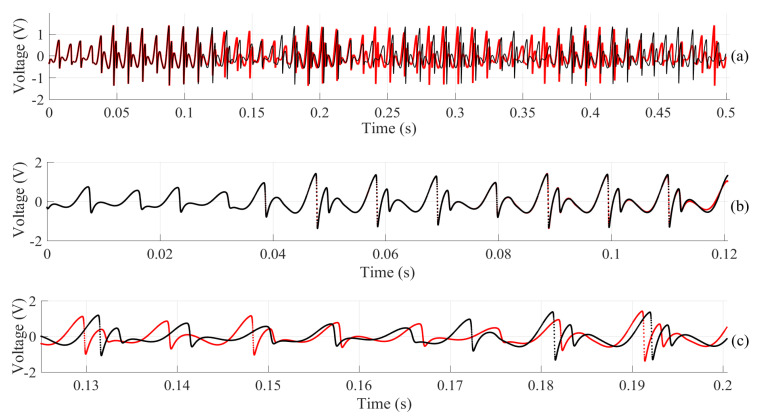
Voltage collected in $\ddot{x}$ for both versions of LTspice software: (
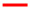
) LTspice IV and (
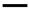
) LTspice XVII. We have shown different intervals of the simulation as follows (**a**) entire simulated time interval: 0 to 0.5 s. (**b**) interval: 0 to 0.12 s. (**c**) interval: 0.12 to 0.2 s. The difference becomes clear after 0.12 s.

**Figure 9 entropy-22-00953-f009:**
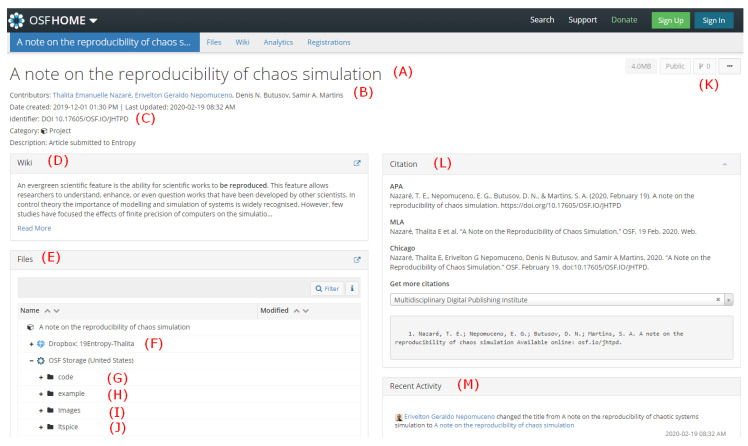
OSF project for this manuscript. It is available on https://osf.io/jhtpd/. Red capital letters are described in the main text.

**Table 1 entropy-22-00953-t001:** NRMSE index result when simulated and experimental data are compared. In this case, all time series (0 to 0.1 s) are compared.

Computer	NRMSE
1	1.4752
2	1.5572
3	1.4841
4	1.4748

**Table 2 entropy-22-00953-t002:** List of computers used for the circuit reproducibility analysis. It is important to stress that the same version of LTspice was used on all these computers.

Computer	System	Configuration
1	Windows 10	Intel Core i5 6200U
2	Windows 10	Intel Dual core 2
3	Windows 8.1	Intel Core i5-3570
4	Windows 8.1	Intel Core i5-4210U

**Table 3 entropy-22-00953-t003:** NRMSE index result for different simulation intervals. The data are divided into windows of 4700 points to observe the evolution of the NRMSE index for each computer. Computer 4 is chosen as a reference since it shows a slightly better performance than Computer 2, the second position candidate.

Computers				
**Iteration**	**1**	**2**	**3**	**4**
1–470	0.6457	1.3960	0.6480	0.6457
1–940	0.9843	1.4110	0.9854	0.9843
1–1410	1.1611	1.4244	1.1616	1.1611
1–1880	1.2396	1.5270	1.2398	1.2396
1–2350	1.2643	1.5132	1.2649	1.2643
1–2820	1.3197	1.5093	1.3194	1.3197
1–3290	1.3797	1.5569	1.3835	1.3797
1–3760	1.3993	1.5453	1.4040	1.3993
1–4230	1.4431	1.5437	1.4567	1.4431
1–4700	1.4752	1.5572	1.4841	1.4748

**Table 4 entropy-22-00953-t004:** NRMSE index result assuming Computer 4 as the reference.

	Computers		
**Iteration**	**1**	**2**	**3**
1–470	0	1.3972	0.0047
1–940	0	1.0699	0.0039
1–1410	0	0.9148	0.0053
1–1880	0	0.9372	0.0108
1–2350	0	1.0563	0.0117
1–2820	0	1.0702	0.0132
1–3290	0	0.9800	0.0241
1–3760	0	1.0075	0.1713
1–4230	0	1.0400	0.3297
1–4700	0.0454	1.0128	0.4415

**Table 5 entropy-22-00953-t005:** Entropy calculation for the signals obtained from Computers 1–4 using LTspice XVII and the practical implementation.

Signal	Entropy (H(x))
Experimental Data	7.3914
Computer 1	7.4008
Computer 2	7.4654
Computer 3	7.3391
Computer 4	7.4004

**Table 6 entropy-22-00953-t006:** Result of the NRMSE index for versions (IV and XVII) of LTspice software.

Iteration	NRMSE
1–2290	0.00037
1–4580	0.00032
1–6870	0.00038
1–9160	0.00181
1–11,450	0.00438
1–13,740	0.00455
1–16,030	0.00451
1–18,320	0.01054
1–20,610	0.02357
1–22,900	0.02644

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
