# Peer review of "A Note on the Reproducibility of Chaos Simulationâ€"

_entropy, 2020, doi:10.3390/e22090953_

Round 1
Reviewer 1 Report
This paper can be accepted in the current version.
Author Response
Dear Reviewer,
Please find attached our reply.
Sincerely,
Dr. E. G. Nepomuceno

Reviewer 2 Report
Authors present a study on the reproducibility of chaos simulations. They compare a Jerk circuit with its simulation in four computers, using the Normalized Root Mean Square Error.
1.- Although authors present the practical implementation of a Jerk circuit in Fig. 1 and Fig. 2, they never show the signals obtained from the circuit.
2.- Initial conditions for simulation are not given. Then, presented simulations are not completely reproducible.
3.- How the authors manipulate the initial conditions for the experimental circuit, such that simulations match with experimental signals?
4.- Authors, in lines 227 and 232 propose using computer 4 as a reference. What is that mean? Signal from computer 4 takes the place of y(t)?
5.- In Table 4, the difference between high and low value is 0.0089. Is this difference really significant to speak about a better performance?
6.- In the same way, in lines 279 and 280, what is "a short reproducibility"? What is the interval to say good reproducibility and poor reproducibility?
7.- If the same data sequence cannot be reproduce, that does not affect the reproducibility like synchronization or control of chaotic systems. Then, what is the relevance of this study?
8.- If a simulation from other work cannot be exactly reproducible, it is factible to reproduce the experimental circuits with almost the same results?
Author Response

(The authors gave the same response as above.)

Reviewer 3 Report
- The authors should review more related works in paragraphs 3 and 4 in the introduction section.
- Section 3 should not only be limited to the related works published in Entropy journal. The discussion of the irreproducibility of numerical simulations and experimental results is good, but can be summarized to around third its current length. The authors should devote part of their discussion on related works to those who discussed reproducibility.
- There are many relevant works, research approaches and advances through the years not mentioned; for example:
- Computer arithmetic, chaos and fractals. Physica D: Nonlinear Phenomena, 1990, 42(1-3), 99-110.
- On the dynamical degradation of digital piecewise linear chaotic maps, 2005, International journal of Bifurcation and Chaos, 15(10), 3119-3151.
- Analyzing logistic map pseudorandom number generators for periodicity induced by finite precision floating-point representation, 2012, Chaos, Solitons & Fractals, 45(3), 238-245.
- Cycle lengths and correlation properties of finite precision chaotic maps, 2014, International Journal of Bifurcation and Chaos, 24(09), 1450107.
- Counteracting the dynamical degradation of digital chaos via hybrid control, 2014, Communications in Nonlinear Science and Numerical Simulation, 19(6), 1970-1984.
- Finite precision logistic map between computational efficiency and accuracy with encryption applications. Complexity, 2017
- Image encryption using finite-precision error, 2019, Chaos, Solitons & Fractals, 123, 69-78.
- Random property enhancement of a 1D chaotic PRNG with finite precision implementation, 2019, Chaos, Solitons & Fractals, 118, 134-144.
- Software and Hardware Implementation Sensitivity of Chaotic Systems and Impact on Encryption Applications. CIRCUITS SYSTEMS AND SIGNAL PROCESSING, 2020
- In the results section, the authors should provide more details on the following:
- “Thus, it is possible to observe that the longer the simulation time, the greater the difference between the results of each computer.” What justifications can the authors provide for this result? Is it related to the nature of chaotic systems and the divergence of nearby trajectories on the long term evolution?
- “With these results, it is possible to demonstrate that the reproducibility of the software is higher when the characteristics of the computers are similar.” Which characteristics (hardware or OS, which hardware component?) really affect the results and cause similarities/dissimilarities?
- Do the results differ on the same computer, but different operating systems or versions of the same software?
What are the Correct Results for the Special Values of the Operands of the Power Operation?, 2016, ACM Transactions on Mathematical Software (TOMS), 42(2), 1-17.
- The authors study the effect of the simulation time/number of iteration, yet, what is the effect of the simulation step? The authors should clearly state its value and whether it can be varied and, if so, its effect on the results.
- The authors should clearly mention the difference between their proposed work and previous related works, e.g., [23]. Observing and reporting the reproducibility problem is not a sufficient contribution. Besides “We have also suggested using the OSF platform as a systematic way to share code, files, and technical details essential to reproduce the results in this paper.”, what are the technical contribution and implications of the obtained results on real life applications?
- The authors should review what other indices were used to report irreproducibility besides NRMSE and justify their choice of it.
- The authors should compare their suggested solutions to the reproducibility problem with the state-of-the-art related works, for example, those mentioned in comment 3).
Author Response

(The authors gave the same response as above.)

Round 2
Reviewer 2 Report
Authors have attended all my comments.
Reviewer 3 Report
Accept
This manuscript is a resubmission of an earlier submission. The following is a list of the peer review reports and author responses from that submission.
Round 1
Reviewer 1 Report
In this work, a case study of reproducibility is presented in the simulation of a chaotic jerk circuit, using the software LTspice. The methodology developed was efficient in identifying the computer with better performance, which allows applying it to other cases in the literature. It seems some valuable.
One point is commented as follow:
The circuit-implemented hardware experiments should be an effective scheme for the reproducibility of chaos simulation. Many references for the circuit-implemented hardware experiments have been reported recently, for examples, Initial-switched boosting bifurcations in 2D hyperchaotic map, Flux-charge analysis of two-memristor-based Chua’s circuit: Dimensionality decreasing model for detecting extreme multistability, and Generating multi-scroll Chua’s attractors via simplified piecewise-linear Chua’s diode. The authors are suggested to briefly review these references.
Reviewer 2 Report
The manuscript employs a chaotic jerk circuit using the software LT spice intending to find a certain reproducibility in four different computers.
In my opinion, the manuscript does not have enough scientific contribution in order to be published as a journal article. In this sense, my recommendation would be extending the manuscript with more detailed analysis.
Moreover, it is not clear how manuscript fits to the scope of Entropy journal.